# A User-Friendly Approach for Routine Histopathological and Morphometric Analysis of Skeletal Muscle Using CellProfiler Software

**DOI:** 10.3390/diagnostics12030561

**Published:** 2022-02-22

**Authors:** Valerio Laghi, Valentina Ricci, Francesca De Santa, Alessio Torcinaro

**Affiliations:** 1Unité Macrophages et Développement de l’Immunité, Institut Pasteur, CNRS UMR3738, 75015 Paris, France; valerio.laghi@pasteur.fr; 2Institute of Biochemistry and Cell Biology (IBBC), National Research Council of Italy (CNR), Via Ercole Ramarini, 32, Monterotondo, 00015 Rome, Italy; valentina_ricci_@hotmail.it (V.R.); francesca.desanta@cnr.it (F.D.S.)

**Keywords:** skeletal muscle, skeletal muscle differentiation, histology, quantitative analysis, myopathies, Duchenne muscular dystrophy, cell localization, histopathological analysis

## Abstract

Adult skeletal muscle is capable of active and efficient differentiation in the event of injury in both physiological and pathological conditions, such as in Duchenne muscular dystrophy (DMD). DMD is characterized by different features, such as continuous cycles of degeneration/regeneration, fiber heterogeneity, chronic inflammation and fibrosis. A well-defined and standardized approach for histological and morphometric analysis of muscle samples is necessary in order to measure and quantify specific regenerative parameters in myopathies. Indeed, non-automatic methods are time-consuming and prone to error. Here, we describe a simple automatized computational approach to quantify muscle parameters with specific pipelines to be run by CellProfiler software in an open-source and well-defined fashion. Our pipelines consist of running image-processing modules in CellProfiler with the aim of quantifying different histopathological muscle hallmarks in *mdx* mice compared to their wild-type littermates. Specifically, we quantified the minimum Feret diameter, centrally nucleated fibers and the number of macrophages, starting from multiple images. Finally, for extracellular matrix quantification, we used Sirius red staining. Collectively, we developed reliable and easy-to-use pipelines that automatically measure parameters of muscle histology, useful for research in myobiology. These findings should simplify and shorten the time needed for the quantification of muscle histological properties, avoiding challenging manual procedures.

## 1. Introduction

Skeletal muscle is a highly dynamic and plastic organ, able to respond to environmental changes and characterized by complete functional recovery upon perturbations such as endurance exercise, overload or muscle injury [1]. These exceptional adaptive features of adult skeletal muscle are reduced or even compromised in conditions such as aging and atrophy or in genetic myopathies, such as Duchenne muscular dystrophy (DMD) [1,2,3]. DMD is a lethal X-linked recessive disease that affects approximately 1/3500 boys and is caused by different mutations in the dystrophin gene, leading to the loss of the functional protein, which is crucial for the proper structure and stability of myofibers [4]. The dystrophin-deficient mouse (C57BL/10ScSn-*DMD^mdx^*/J), referred to as *mdx* mouse, represents the most frequently used animal model to study DMD, although the pathology is less severe in this animal compared to DMD patients [5,6]. In both cases, this fatal myopathy leads to continuous cycles of degeneration and regeneration, resulting in high heterogeneity in fiber size and distribution as well as an increase in centrally nucleated fibers (CNFs) [7]. Another key feature of DMD is chronic inflammation, resulting in persistent inflammatory cell infiltration, mainly macrophages, upon the degeneration of myofibers, accompanied by irreversible extracellular matrix deposition (ECM), leading to fibrosis [8,9].

The study of skeletal muscle physiology or pathologies mainly relies on histological analyses of muscle cross-sections. This analysis is commonly carried out by measuring the cross-sectional area (CSA) or minimum Feret diameter (MFD) of myofibers and the fiber size distribution in order to evaluate muscle fiber size and heterogeneity within the muscle [10]. Indeed, in physiological conditions, wild-type mice show a homogenous fiber size distribution and a constant CSA in the absence of perturbations. On the contrary, *mdx* mice usually show decreased CSA and an increase in fibers with a smaller caliber, together with high fiber heterogeneity, which becomes more evident with the progression of the disease. Notably, MFD quantification is usually preferable to CSA [11]. CSA and fiber size distribution measurements are usually performed by anti-laminin immunofluorescence with the goal of detecting fiber boundaries, whereas, for CNF quantification, nuclei detection is also necessary, which can be accomplished, for example, using 4′,6-diamidino-2-phenylindole (DAPI) [1]. This type of analysis can be performed by many software packages that can allow either manual or automated quantification, although both of these procedures have crucial pros and cons. Indeed, although manual quantification accounts for the critical assessment of the investigator concerning the biological problems under examination, this approach is undoubtedly time-consuming and highly subjective among users. On the contrary, automatized software packages are designed to save time and to standardize the procedure but often do not include a step of “manual revision” by the user, thus compromising the accuracy of the quantification. Moreover, some of these software platforms are not open-source, can be difficult to implement and require specific operating systems or a good knowledge of programming languages [10,12,13,14,15,16,17,18]. Additionally, automatized software is usually not designed for the quantification of a cell population or the extracellular matrix deposition within the muscle, two fundamental features in myobiology [1,17]. Different readily available software packages are often optimized towards one or more parameters, reducing the ability of the user to mine different data measurements and the versatility of the software [19].

CellProfiler represents a robust, user-friendly and open-access software platform with algorithms and features that facilitate high-throughput work in biological research [20]. Advanced algorithms for image analysis are organized in individual modules that can be inserted in a sequential order to generate a customizable pipeline to identify or measure biological elements, named “objects”, or quantify positive areas in acquired images [20,21,22].

In the current manuscript, we propose a method to perform image analysis of muscle sections by using pipelines built with CellProfiler software, which have been recently implemented and updated to CellProfiler 4 [23]. Specifically, we present the data obtained by using a pipeline, which we named MyoProfiler, to measure MFD, CNF, PNF, cell localization and the number of macrophages in muscle sections from *mdx* mice compared with wild-type ones. We also developed another pipeline, which we named SiriusProfiler, for the precise quantification of extracellular matrix deposition. Moreover, with the goal of validating the performance of our method, we compared automatic quantification, performed using CellProfiler, with manual quantification, performed using Fiji software. The results show that these pipelines allow the automatic analysis of multiple images in a quick and reliable manner by using a single software package for multiple outputs, thus representing useful tools for the quantification of key muscle parameters in both physiological and non-physiological conditions.

## 2. Materials and Methods

### 2.1. Mice and Ethical Approval

Wild-type (C57BL/10J, The Jackson Laboratory, Bar Harbor, ME, USA) and dystrophic *mdx* mice (C57BL/10ScSn-*DMD^mdx^*/J, The Jackson Laboratory, Bar Harbor, ME, USA) were purchased from Charles River. Five-month-old wild-type and *mdx* mice were used for experiments. All experimental protocols and procedures were conducted following the National Ethical Guidelines (Italian Ministry of Health; D.L. 26, 4 March 2014), approved by the local ethics committee (protocol number 375/2019/PR). Animals were housed at controlled temperature (22 ± 1 °C) and humidity (60 ± 5%) and maintained under a 12 h/12 h light/dark cycle with ad libitum access to food and water.

Mice were euthanized and then dissected in order to carefully excise tibialis anterior (TA) muscles from the hind limbs. Collected TA muscles were mounted in Optimal Cutting Temperature (OCT, Tissue Tek^®^, Sakura Finetek, Alphen aan den Rijn, The Netherlands, Europe) compound and then frozen in liquid nitrogen-cooled isopentane (2-methylbutane; Sigma-Aldrich, Merck KGaA, Burlington, MA, USA). Embedded muscles were then cross-sectioned at a thickness of 8 µm using a Leica cryostat (Leica CM1850UV, Wetzlar, Germany) set at −25 °C, and sections were stored in a −80 °C freezer.

### 2.2. Immunofluorescence of Muscle Sections and Image Acquisition

The immunofluorescence of muscle sections was performed following a previously described procedure [3,24]. Primary antibodies used for this study were rabbit polyclonal antibody raised against laminin, α1 (Sigma-Aldrich, Merck KGaA, Burlington, MA, USA; Cat#: L9393, RRID:AB_477163, 1:500) and rat monoclonal antibody raised against F4/80 (Bio-Rad Laboratories, Hercules, CA, USA; Cat#: MCA497G, RRID:AB_872005, 1:300). Secondary antibodies for immunofluorescence were Alexa Fluor^®^ 488 goat anti-rabbit IgG (H+L; Thermo Fisher Scientific, Waltham, MA, USA; Cat#: A11034, RRID:AB_2576217, 1:500) and Alexa Fluor^®^ 594 goat anti-rat IgG (H+L; Molecular Probes, Eugene, OR, USA; Cat#: A11007, RRID:AB_141374, 1:500). Nuclei were counterstained with 4′,6-diamidino-2-phenylindole (DAPI; Thermo Fisher Scientific, Waltham, MA, USA; Cat#: D1306, RRID:AB_2629482).

Representative images of TA muscle immunofluorescences were acquired using an Olympus confocal microscope (Olympus FV1200, Olympus, Tokyo, Japan) with 40× magnification and visualized with FV10-ASW software (version 4.2; Olympus, Tokyo, Japan). Images for histological analysis were acquired using an Olympus BX53 microscope mounting an XM10 cam (Olympus, Tokyo, Japan) and using “cellSens Standard” software (version 1.17; Olympus, Tokyo, Japan). We acquired adjacent images at 10× magnification of the whole muscle section from both WT and *mdx* mice. A few fields with evident histological defects were removed before the analysis in order to avoid artifacts. Images were saved and exported as 16-bit images (grayscale images).

### 2.3. Staining for Extracellular Matrix Deposition and Image Acquisition

Sirius red staining is commonly used to detect extracellular matrix deposition and fibrosis within tissue sections. Briefly, muscle cryosections were thawed and then fixed with Bouin’s solution (Sigma-Aldrich, Merk KGaA, Burlington, MA, USA; Cat#: HT10132) for 1 h, washed and then stained with Picrosirius red dye (Direct Red 80; Sigma-Aldrich, Merk KGaA, Burlington, MA, USA; Cat#: CI 35780) for 1 h, followed by sequential dehydration in 90%, 100% ethanol and xylene and then mounted with EUKITT (Sigma-Aldrich, Merk KGaA, Burlington, MA, USA; Cat#: 03989). Images were acquired using an Olympus BX-41 microscope (Olympus, Tokyo, Japan) with 10× magnification and visualized using “cellSens Entry” software (version 3.1.1, Olympus, Tokyo, Japan). Specifically, we acquired adjacent images at 10× magnification of entire muscle sections from both genotypes.

### 2.4. CellProfiler-Based Pipelines for Muscle Analysis

CellProfiler, developed by the Carpenter Lab at the Broad Institute of Harvard and MIT, is open-access software and available for Windows and macOS [20,21]. CellProfiler code was written using Python [22], and an updated, faster version of CellProfiler was recently released (CellProfiler 4) [23]. Java (www.java.com, accessed on 20 October 2018) installation and update are required prior to CellProfiler installation. Inexperienced users are encouraged to read the CellProfiler manual before using it. For the analysis of data described in this paper, we used the latest version (4.2.1) of CellProfiler downloaded from the official CellProfiler website (www.cellprofiler.org, accessed on 11 October 2021) and installed it on a laptop computer (Intel^®^ Core™ i7 4500 U CPU @1.80 GHz 2.40 GHz, 8.00 GB RAM, and 64 bit Windows 10 Home operating system). CellProfiler can process a wide range of image formats using the BioFormats library (complete list of formats permissible here https://docs.openmicroscopy.org/bio-formats/5.9.2/supported-formats.html, accessed on 20 February 2018).

To use a pipeline, the user has to run CellProfiler (version 4.2.1), go to “file”, select “open project” and run the project corresponding to the pipeline of interest. Alternatively, .cpproj or .cppipe files can be run. Then, a list of images can be dropped into the Images module. Image processing and data extraction can be performed through Metadata, NamesAndTypes and Groups modules: for each module, a caption with detailed information is available. Together with these four standard modules, custom modules are displayed as soon as the pipeline/project is opened. We also added captions for each module of both pipelines that we designed. The shared workflow proceeds with data processing, including the pipeline of interest (composed of defined modules), and then with the test mode (Start Test Mode) in order to check the result of each module, followed by image analysis (Analyze Images). Test mode is particularly convenient when the user is designing a new pipeline or implementing an old one in order to check how the pipeline itself works with different image sets. Before running these commands, modules can also be selected/deselected (checkmark) or hidden, depending on the outputs that have to be displayed. The analysis ends with the generation of output data and a spreadsheet. It is important to define input and output folders before image processing. Pipelines developed in our lab are available in the Appendix A). Additionally, in the Appendix A section, we provide a troubleshooting guide (Troubleshooting_guide, Appendix A).

### 2.5. Validation of CellProfiler-Based Pipelines by Fiji

Quantifications obtained with pipelines designed with CellProfiler were validated using Fiji software [25,26]. The quantification of the minimum Feret diameter (MFD) of muscle fibers began using anti-laminin, α1-labeled images. DAPI-labeled nuclei and anti-F480-stained macrophages were manually counted using the Cell Counter plugin. CNFs were quantified by combining anti-laminin, α1-labeled images and DAPI-labeled images. All numerical data were exported to Excel files and used for final quantifications. Sirius red quantifications with Fiji were performed using the Color Deconvolution plugin, and the red image was thresholded using the Otsu threshold method [27]. All analyzed images and samples used for quantifications in CellProfiler were used for the validation with Fiji.

### 2.6. Statistical Analysis

Data are presented as means plus/minus standard error of the mean (SEM). Output data were compared by 2-tailed unpaired Student’s *t*-test. Results with *p* value < 0.05 were considered statistically significant. All “*p* values” are indicated on the graphs in the figures. All data analyses were performed using GraphPad Prism 9.3 (GraphPad Software, San Diego, CA, USA).

## 3. Results

### 3.1. Fully Automated Segmentation of Muscle Fibers by MyoProfiler Pipeline

Experiments concerning skeletal muscle regeneration usually require an overview of muscle architecture, which can be evaluated, for instance, by immunostaining muscle cross-sections with antibodies that recognize the sarcolemma (e.g., caveolin) or the basal lamina (e.g., laminin, α1) of muscle fibers [28]. In our study, we stained tibialis anterior (TA) muscle cryosections of both wild-type (WT) and *mdx* mice with anti-laminin, α1 (muscle fiber boundaries) and anti-F480 (macrophage surface marker), and nuclei were counterstained with 4′,6-diamidino-2-phenylindole (DAPI). We used 16-bit grayscale images as input. If the input images are RGB color images, it is necessary to add a ColorToGray module for each fluorophore.

For the quantification of minimum Feret diameter (MFD), we set up a pipeline primarily aimed at better defining the edges of myofibers. The workflow of the first part of the MyoProfiler pipeline is reported in Figure 1A and downloadable in the Appendix A). The first module of the pipeline was RescaleIntensity, which stretches the intensity of the image to the full intensity range, which is particularly useful for 16-bit images (Figure 1B). The next steps aimed at making the boundaries of muscle fibers sharper, and to this end, we set up a custom “unsharp mask”. Briefly, we used GaussianFilter to blur the input image and reduce its background noise (Figure 1C). We then used two ImageMath modules, which perform simple mathematical operations on image intensities, as follows: with the first one, we subtracted the blurred image, obtained with GaussianFilter, from the rescaled laminin image in order to sharpen the edges of muscle fibers; secondly, we added the sharpened laminin image to the original one (Figure 1C, ImageMath 1+2).

As the name implies, the EnhanceOrSuppressFeatures module is designed to enhance or suppress specific image features of interest. In this case, we enhanced “Line structures” from original rescaled images in order to recover and highlight low-intensity linear structures (Figure 1D, EnhanceFeatures 1). With another EnhanceOrSuppressFeatures module, we made muscle fiber boundaries sharper and clearer by enhancing the “Neurites” feature and using “Tubeness” as an enhancing method (Figure 1D, EnhanceFeatures 2). The signal of the output image was then enhanced using an ImageMath module. Afterwards, another ImageMath module took the average of the two enhanced laminin output images (“Enhance_Line_Laminin_09” and “Enhance_Laminin_11”) to further reconstruct the laminin signal without increasing the background noise (Figure 1D, ImageMath 3+4). A subsequent Closing module, which applies a Dilate/Erode cycle, closed the intensity gaps between pixels with a disk-shaped structuring element of 2 pixels in order to connect interrupted fiber boundaries as much as possible (Figure 1E). We observed that increasing the size of the structuring element could lead to the formation of small oversegmented fibers. Afterwards, we used the MedianFilter module to reduce the salt-and-pepper noise in the image while still preserving the positive signal. (Figure 1F).

The Threshold module is necessary to detect the entire positive signal before the segmentation of muscle fibers. We selected a global threshold strategy and Otsu [27] with three classes as a thresholding method, since the percentage of the image covered by foreground varied from image to image, especially in *mdx* samples. Accordingly, we chose to assign the middle-intensity class to the background in order to exclusively select the true-positive signal. The output image is a binary image in which the negative signal is set to 0, while the positive signal is set to 1 (Figure 1G). In order to ensure the correct performance of the segmentation, we also added a Morph module, which further closed the gaps between muscle fibers and filled small holes, such as capillaries and nerve bundles (Figure 1H). We then inverted the pixel values of the binary input image by using an ImageMath module (with the “Invert” operation). This step is necessary to ensure that the muscle fibers are segmented correctly, as explained in the next step (Figure 1I, ImageMath 5).
Figure 1MyoProfiler: detection and segmentation of myofibers using anti-laminin, α1-stained muscle cross-sections. (**A**) Workflow of the first part of MyoProfiler pipeline. (**B**) RescaleIntensity module for laminin signal. Scale bar = 100 µm. (**C**) Custom “unsharp mask” step (GaussianFilter and ImageMath 1+2) for sharpening myofiber boundaries. (**D**) Line structures and myofiber boundaries are further enhanced with 4 sequential modules (two EnhanceAndSuppressFeatures modules and two ImageMath modules). (**E**) Closing module closes the intensity gaps between pixels. (**F**) MedianFilter module reduces salt-and-pepper background noise. (**G**) Threshold module detects positive signal and produces a binary image. (**H**) Morph module closes the gaps between muscle fibers and fills small holes. (**I**) Inverted binary image (ImageMath 5). (**J**) Color map image of segmented myofibers with IdentifyPrimaryObjects module. FilterObjects module discards wrong muscle fibers (magenta arrows point to discarded fibers). (**K**) OverlayOutlined generated an output image in which both segmented (red) and discarded (yellow outlines and pointed by cyan arrows) myofibers are outlined.
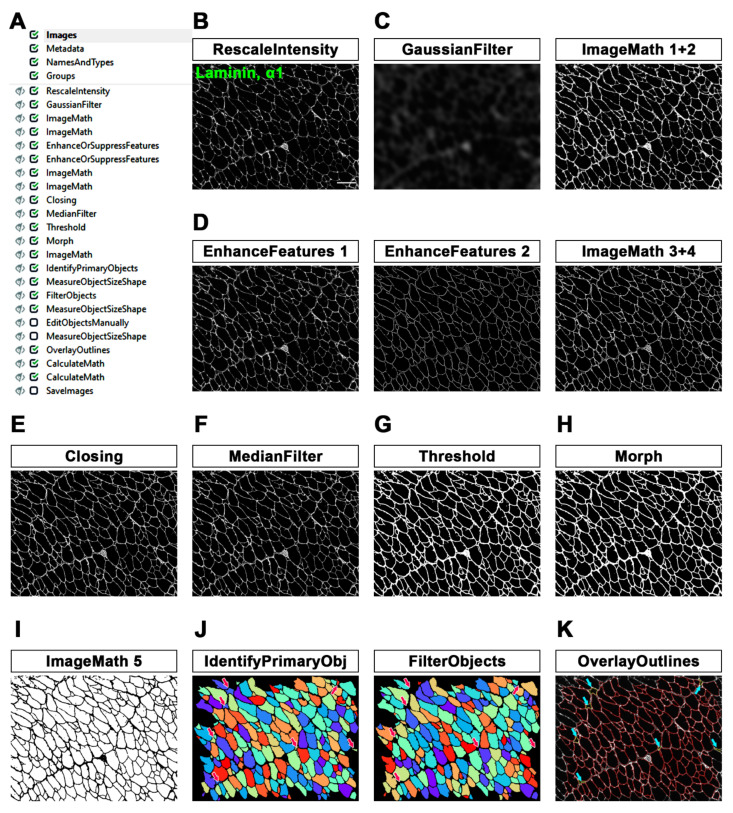


The IdentifyPrimaryObjects module received the inverted thresholded image as input and identified muscle fibers as objects by using a global minimum cross-entropy threshold [29]. Objects touching borders were discarded. As a method to distinguish and segment clumped objects, we used “Intensity” (used for objects that are dimmer towards their edges and usually brighter in the middle) and “Shape” as the method to draw dividing lines between clumped objects. This approach follows the indentation of two touching objects, which is useful for dividing fused muscle fibers. Another fundamental parameter to be defined is the typical diameter range (in pixel units) of objects. We found that a range of 18–2000 pixels in diameter was suitable for our images (Figure 1J). However, it is possible to measure the diameter of objects from the Images module before processing the image set. Afterwards, we used the MeasureObjectSizeShape module to measure the area and shape features of identified muscle fibers. FilterObjects (followed by a MeasureObjectSizeShape module) is a very important module, since it allows the exclusion of objects that do not meet the criteria for specific parameters determined by the user. For our pipeline, we decided to discard all objects below 40 pixels in area and objects with a minimum value of 0.32 for the form factor and a minimum of 0.75 for solidity (Figure 1J; magenta-colored arrows point to discarded objects). Nevertheless, we found that a minimal number of elongated muscle fibers (which is rarely present only in *mdx* mice or in poor histological sections) could be discarded. Overall, this setting was solid and successful in all image sets used for this paper. We also decided to add an optional module (EditObjectsManually #22, followed by a MeasureObjectSizeShape #23 module), which can be enabled and used in the analysis. Once the analysis of an image set reaches the EditObjectsManually module, an editing user interface allows objects to be created, removed and edited. The interface shows the current image overlaid with colored outlines of the selected objects. A number of operations are available: remove an object, restore a removed object, edit objects, finish editing an object and abandon changes to an object. Moreover, quick key commands for object editing are listed. Object editing includes deletion, manually drawing, joining and splitting, or removal of an object. Furthermore, it is necessary to replace the input “Muscle_Fibers_20” of modules 24, 25, 26, 33 and 38 with “Edited_Muscle_Fibers_22”. In “Select measurement” and “data to export” options of the ExportToSpreadsheet module, it is necessary to select/indicate the correct modules accordingly. However, this semi-automatic step can be skipped if the overall detection and segmentation of muscle fibers have been performed well. However, we suggest always using the “Test Mode” and the “Show display” mode provided by CellProfiler before analyzing new image sets.

The output of all of these steps can be visualized with the OverlayOutlines module, with which the boundaries of final muscle fibers (from FilterObjects module) are identified (red) and displayed on the original image. Discarded objects are outlined in yellow in Figure 1K and indicated with cyan-colored arrows. Finally, the two following CalculateMath modules were used to convert the calculated CSA of muscle fibers from pixel to μm^2^ and the MFD from pixel to μm. Conversion factors for both modules were calculated starting from the pixel/μm ratio assigned to the immunofluorescence images used for this procedure. Finally, we also developed an alternative version of MyoProfiler (MyoProfiler_variant, Appendix A), in which we selected an adaptive three-class Otsu threshold with an adaptive window set to 500 pixels. This could be useful for an input image if there is uneven intensity across the image.

### 3.2. Quantification of Nuclei, CNFs and PNFs in Muscle Sections

A well-known hallmark of skeletal muscle regeneration in physiological and pathological conditions (e.g., DMD) is the presence of centrally nucleated fibers (CNFs), corresponding to foci of regeneration in the injured muscle. This fact represents a reliable aspect of muscle condition, especially in *mdx* mice in which muscle fibers remain centrally nucleated even at the end of the regenerative process, whereas the nuclei of regenerated healthy muscles migrate towards the periphery of myofibers (i.e., peripherally nucleated fibers or PNFs) [17,30]. Therefore, we co-stained tibialis anterior (TA) muscle cryosections of both wild-type and *mdx* mice with laminin, α1 and with 4′,6-diamidino-2-phenylindole (DAPI).

The workflow of the second part of the MyoProfiler pipeline is shown in Figure 2A. First, we used RescaleIntensity on the DAPI signal (Figure 2B), similarly to what was previously performed with laminin. We then used the MedianFilter module to reduce the background noise, followed by the EnhanceOrSuppressFeatures module (Figure 2C, MedianFilt+Enhance). In this case, we were interested in enhancing the speckles of nuclei, since the DAPI signal is usually characterized by spots of enhanced intensity relative to the background. The module enhances speckles using a white tophat filter. The feature size was set to 10 pixels, which corresponds, in our images, to the typical nucleus diameter. Nuclei were segmented using IdentifyPrimaryObjects, setting a diameter range of 6–20 pixels and using the minimum cross-entropy threshold method [29] with an adaptive window of 20 pixels in size, corresponding to the maximum diameter set for nuclei. Objects touching borders were discarded. Intensity was used as a method to both distinguish clumped objects and to draw lines between them (Figure 2D, IdentifyPrimaryObj). The OverlayOutlines module outlines identified nuclei to the rescaled original DAPI image (yellow outlines; Figure 2E).

Once we had successfully identified and segmented both muscle fibers and nuclei, we focused on CNF and PNF detection. To this end, we used the ExpandOrShrinkObjects module to shrink identified muscle fibers by 5 pixels. This procedure is necessary to exclude all peripheral nuclei juxtaposed to myofiber boundaries (Figure 2F, ExpandOrShrinkObj). With another ExpandOrShrinkObjects module, this time, we shrank the nuclei to one point in order to clearly mask and detect nuclei of CNFs in the following steps. As previously mentioned, we took advantage of the MaskObjects module to mask and remove all one-point nuclei outside shrunken muscle fibers, thus considering only nuclei inside myofibers (Figure 2G, Shrink+MaskObjects). Afterwards, the RelateObjects module assigned the relationship between previously identified objects. In this case, “child objects” are all objects (nuclei) inside “parent objects” (shrunken fibers). Shrunken fibers were then classified into CNFs and PNFs by the ClassifyObjects module. Interestingly, as output values, it is also possible to quantify the number of nuclei per CNF. Finally, we added an OverlayOutlines to highlight myofiber boundaries (red), nuclei (yellow), shrunken fibers (cyan) and nuclei of CNFs (outlined as pink dots; Figure 2G, RelateObj+Overlay).
Figure 2MyoProfiler: detection of centrally nucleated fibers (CNFs) and segmentation of nuclei and macrophages. (**A**) Workflow of the second part of MyoProfiler pipeline. (**B**) RescaleIntensity module for DAPI signal. Scale bar = 100 µm. (**C**) MedianFilter module reduces salt-and-pepper background noise and EnhanceOrSuppressFeatures enhances the “speckles” signal of nuclei. (**D**) Color map image of segmented nuclei generated with IdentifyPrimaryObjects module. (**E**) OverlayOutlines generates an output image in which nuclei are outlined (yellow). (**F**) Color map image of shrunken myofibers generated with ExpandOrShrinkObjects module. (**G**) Output image generated from the shrinking of nuclei to one pixel and masking with shrunken myofibers (Shrink+MaskObjects) for the identification of CNFs. OverlayOutlines outlines segmented (red) and shrunken myofibers (cyan), as well as nuclei inside (dark pink dots) and outside (yellow) of shrunken fibers. (**H**) Workflow of the third part of MyoProfiler pipeline. (**I**) RescaleIntensity module for F4/80 (macrophage) signal. Scale bar = 100 µm. (**J**) MedianFilter module reduces salt-and-pepper background noise, and EnhanceOrSuppressFeatures enhances the “speckles” signal of macrophages. (**K**) Color map image of segmented macrophages generated with IdentifyPrimaryObjects module. (**L**) ExpandOrShrinkObjects module enlarges nuclei to improve detection of macrophages. (**M**) Color map image of nuclei belonging to macrophages (MaskObjects module); other nuclei are outlined in purple. RelateObjects module assigns the relationship between nuclei and macrophages. (**N**) OverlayOutlines outlines nuclei belonging (yellow) and not (purple) to macrophages. Macrophage boundaries are outlined in cyan.
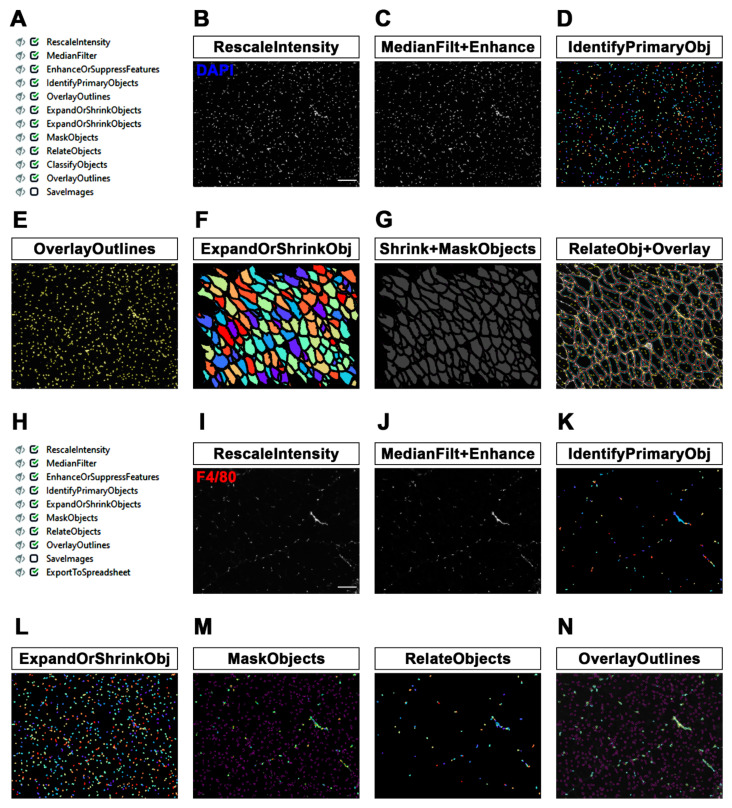


### 3.3. Detection and Quantification of Macrophages in Skeletal Muscle

Cell localization is an important aspect in the field of cell biology and imaging; it is intended to locate and analyze specific cell populations within a tissue or organ. Usually, a specific cell population is identified by using cells expressing a reporter gene encoding for a fluorescent protein (e.g., GFP) or by using an antibody directed towards a protein specific for the cell population. For both methods, counterstaining with a nuclear dye (e.g., DAPI) is suggested [31]. In the context of skeletal muscle histology, cell localization is particularly important for the detection of muscle stem cells (i.e., satellite cells), interstitial cells (e.g., fibroadipogenic progenitors) and infiltrating immune cells, such as monocytes/macrophages [3,30].

In this study, we detected and quantified the number of macrophages in muscle sections by automatically counting the number of cells (DAPI) co-stained with an antibody raised against F4/80, a pan-macrophage surface marker. Specifically, in the third part of the MyoProfiler pipeline (Figure 2H), we applied the RescaleIntensity module to an original F4/80 image (Figure 2I), followed by MedianFilter for removing salt-and-pepper noise and the EnhanceOrSuppressFeatures module. This module enhances speckles using a white tophat filter. We chose a feature size of 100 pixels, since macrophages can assume different shapes, reflecting the position occupied by macrophages within the tissue in vivo with respect to an in vitro culture. Indeed, macrophages can also assume an elongated shape inside a tissue (Figure 2J, MedianFilt+Enhance).

We then took advantage of the IdentifyPrimaryObjects module to detect macrophages as F4/80-positive signals. We chose the minimum cross-entropy threshold method, with an adaptive window of 100 pixels in size; as the diameter range, we chose 8–100 pixels. Objects touching borders were discarded, as usual. We used intensity to distinguish clumped objects and shape to draw the dividing line between clumped objects, since the peak intensity of these objects is more variable than that of the nuclei. Moreover, we selected the “Log transform before thresholding” option, which helps to detect areas of staining that have a wide dynamic range, such as the F4/80 signal (Figure 2K, IdentifyPrimaryObj). Afterwards, we decided to use an ExpandOrShrinkObjects module to expand the area of the nuclei by 2 pixels in order to better detect macrophages. Indeed, F4/80 is a membrane marker, and this module prevents objects with the DAPI signal juxtaposed to the F4/80 signal from being discarded (Figure 2L, ExpandOrShrinkObj). With the MaskObjects module, we were able to mask (remove) objects or regions outside the region of interest. The objects that were partially masked were removed on the basis of the overlap fraction. Mask objects (macrophages) will keep an object (expanded nuclei generated by the previous module) only if the overlap is at least 0.1, meaning that 1/10 of an object must be in the masking region (expanded nuclei are outlined in purple; Figure 2M, MaskObjects). Afterwards, the RelateObjects module assigned the relationship between previously identified objects (Figure 2M, RelateObjects). In this case, “child objects” are all objects (nuclei) inside “parent objects” (macrophages). Finally, we used the OverlayOutlines module to outline expanded nuclei (purple), masked macrophage nuclei (yellow) and macrophage boundaries (cyan; Figure 2N). Interestingly, the quantification of macrophages (F4/80-positive cells) is an approach that is potentially applicable to all quantification methods involving the co-localization or co-staining of DAPI with another fluorescent dye or antibody, which is useful for the detection of many cell populations residing in tissues and organs. Indeed, it can be applied to other cell populations of skeletal muscle, such as satellite cells and fibro-adipogenic progenitors (Section A.1), or to cell populations of other tissues and organs.

### 3.4. Comparison between CellProfiler-Based Fully Automatic Quantification and Non-Automatic Quantification

In order to validate the accuracy of the MyoProfiler pipeline developed in our lab, we decided to compare output data generated with CellProfiler with those generated with Fiji, a widely used software package for image analysis. To this end, we used images acquired from the immunofluorescence of TA muscle from wild-type (WT) and *mdx* mice stained for laminin, α1 (green), DAPI (blue) and F4/80 (red). Figure 3A shows a compact muscle architecture in WT and a heterogeneous myofiber composition in *mdx* muscle, together with a high increase in CNFs and massive macrophage infiltration (Figure 3A).

The first output data generated by the MyoProfiler analysis was the MFD of muscle fibers, a well-known parameter for quantifying fiber size. Our data showed that MFD values quantified with MyoProfiler were extremely similar to those obtained with Fiji (WT: *p* = 0.8117; *mdx*: *p* = 0.8188). Consistently, fiber size distributions were also almost equal when comparing the two approaches (Figure 3B). Moreover, the numbers of detected and segmented nuclei were also comparable between the two methods (WT: *p* = 0.4880; *mdx*: *p* = 0.5724; Figure 3C). Consistency in detecting nuclei and segmenting myofibers was also observed for the quantification of CNFs (WT: *p* = 0.8906; *mdx*: *p* = 0.7162) and PNFs (WT: *p* = 0.9413; *mdx*: *p* = 0.6020; Figure 3D). Interestingly, MyoProfiler also allowed the quantification of the number of nuclei per myofiber. As expected, we observed a significant increase in the number of nuclei per myofiber in *mdx* mice compared to their WT littermates (Figure 3E). Finally, we found that MyoProfiler was also proficient in segmenting macrophages in skeletal muscle, despite the irregular F4/80 signal (WT: *p* = 0.6536; *mdx*: *p* = 0.8367; Figure 3F). Collectively, these data demonstrated that the MyoProfiler pipeline efficiently and robustly quantified many fundamental parameters for routine muscle analysis.

We calculated an average processing time of 10 min for quantifying 10 image sets (each image set composed of laminin, α1, DAPI and F4/80 images) with MyoProfiler. Conversely, quantification performed with Fiji on the same image sets took several hours. Overall, our pipeline represents a robust, reliable and fast approach for quantifying many histological features starting from immunofluorescence images of muscle sections.
Figure 3Quantitative measurements automatically performed with MyoProfiler and compared with Fiji software. (**A**) Representative immunofluorescence images of tibialis anterior sections from wild-type (WT) and *mdx* mice. Sections were stained with anti-laminin, α1 (green) and anti-F4/80 (red) antibodies and counterstained with DAPI (blue). Scale bar = 50 µm. (**B**) Minimum Feret diameter (MFD) quantification and fiber size distribution of myofibers. (**C**) Quantification of the number of nuclei. (**D**) Quantification of centrally nucleated (CNFs) and peripherally nucleated fibers (PNF). (**E**) Quantification of the number of nuclei per CNF performed by CellProfiler. (**F**) Quantification of the percentage of macrophages. Data are expressed as mean ± SEM, and unpaired *t*-test was used for comparison (N = 3 for WT and N = 4 for *mdx*; * = *p* < 0.05; ** = *p* < 0.01; *** = *p* < 0.001).
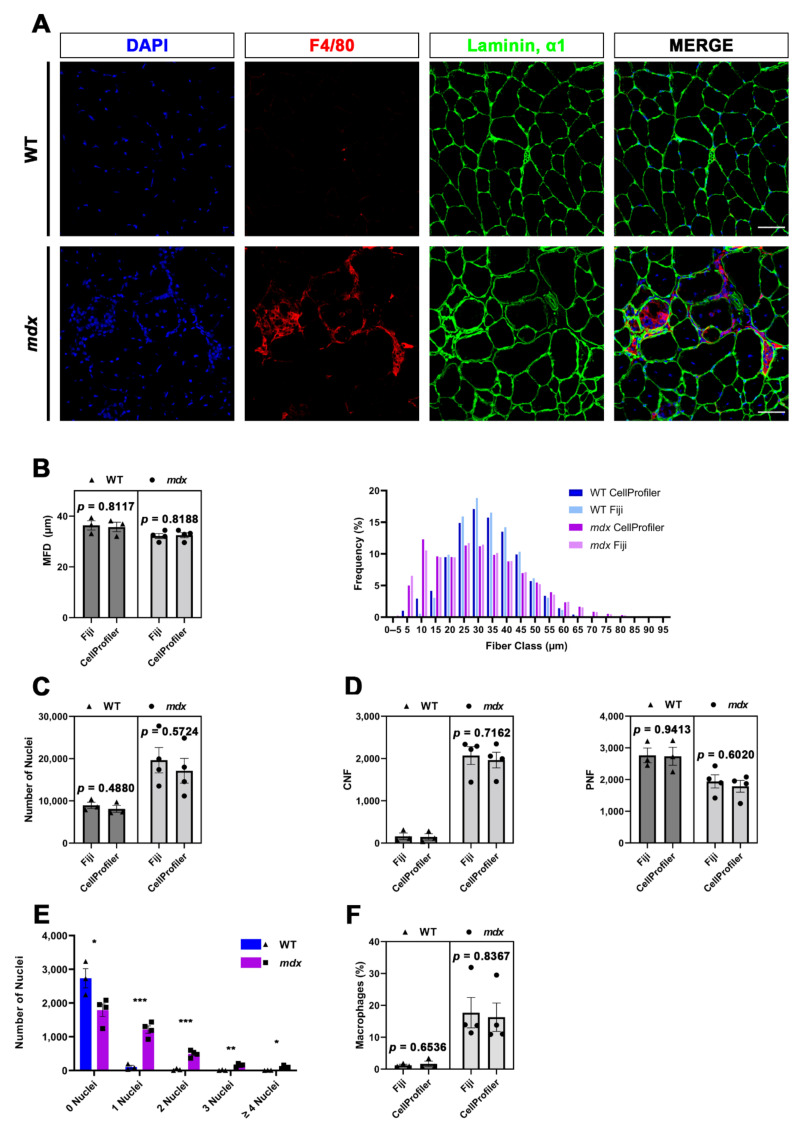


### 3.5. Quantification of Extracellular Matrix in Skeletal Muscle

Picrosirius red staining is a fast and reliable staining method that is largely used to identify and quantify the level of extracellular matrix (ECM) deposition in healthy and diseased muscle cross-sections. Direct Red 80 is a dye that stains the ECM red, whereas the cytoplasm and muscle fibers are yellow.

The pipeline that we created for ECM quantification (SiriusProfiler, Appendix A) comprises fewer modules (Figure 4A) with respect to the previous one. Once we updated the image set corresponding to a muscle section, we assigned the name “Sirius” to all of the images of the uploaded image set. We used RGB colored images (Figure 4B), so we selected “Color image” from the drop-down menu. We decided to add two optional sequential pre-processing modules designed to correct the background of the image when the illumination is not consistent. The illumination across the image is first calculated (CorrectIlluminationCalculate) and then fixed accordingly (CorrectIlluminationApply).

The UnmixColors module creates different grayscale images starting from the original colored one. The main goal is to select the best absorbance for Sirius red staining. Specifically, we split the image into blue, yellow and red by selecting Fast blue (blue), DAB (yellow) and Fast red (red). The reliability of this approach is clear, since the grayscale image resulting from the Fast red image perfectly corresponds to the red area of the original image. Indeed, the grayscale Fast blue image is almost completely dark, while the greyscale DAB image highlights cytoplasm and muscle fibers (Figure 4C).

The following module, Threshold, is the most critical when dealing with the quantification of areas. We applied a global Otsu thresholding method since it allows the division of a grayscale input image into three classes. Indeed, the intensity of Sirius red appeared to be composed of high- and middle-intensity signals. Therefore, we assigned pixels in the middle-intensity class to the foreground (Figure 4D). We also added two optional modules to perform a quality check of the quantification. Briefly, the thresholded image is converted into objects (ConvertImageToObjects), and then this output image is used to display the outlines (in red) of the selected area in the original image.

The positive area is then quantified and multiplied by a conversion factor (0.2289), calculated starting from the pixel/μm ratio value, in order to convert the calculated area from pixel to μm^2^. Output data are generated as a “.txt” file. Finally, we added another two optional modules, which allowed us to save the Fast red and thresholded image.

Representative images of whole tibialis anterior muscles from WT and *mdx* mice stained with Sirius red clearly show a massive increase in fibrotic scars in *mdx* mice, a typical hallmark of DMD (Figure 4E). We calculated an average processing time of 80 s for quantifying 10 images with SiriusProfiler. Conversely, manual quantification with Fiji took approximately 13 min for 10 input images. As expected, the quantification of the percentage of the positive area increased in *mdx* mice compared to their WT littermates. Notably, the quantification performed with CellProfiler was almost identical to the one performed with Fiji (WT: *p* = 0.7361; *mdx*: *p* = 0.9591; Figure 4F) in both genotypes, thus validating the robustness of our fully automatic pipeline for ECM quantification.

Among other applications, SiriusProfiler pipeline, allows also to quantify ECM, with appropriate modifications, by WGA-stained sections (Section A.2) or immunofluorescence of muscle sections for markers specific for ECM (e.g., anti-Collagen I; Section A.2). Finally, SiriusProfiler pipeline can quantify ECM also in Masson’s Trichrome-stained muscle sections (Section A.3).
Figure 4SiriusProfiler workflow and quantitative measurements. (**A**) Workflow of SiriusProfiler pipeline. (**B**) Original input Sirius red image uploaded in Images module. Scale bar = 200 µm. (**C**) UnmixColors module for the split of input image into red (Fast red), yellow (DAB) and blue (Fast blue) components. (**D**) Threshold image of the red signal. (**E**) Representative images of Sirius red-stained whole muscle sections of tibialis anterior muscle from wild-type (WT) and *mdx* mice. Scale bar = 500 µm. (**F**) Percentage of collagen-positive area (Sirius red staining) in WT and *mdx* mice, quantified with CellProfiler (SiriusProfiler) and Fiji software. Data are expressed as mean ± SEM, and unpaired *t-*test was used for comparison (N = 3 for WT and N = 4 for *mdx*).
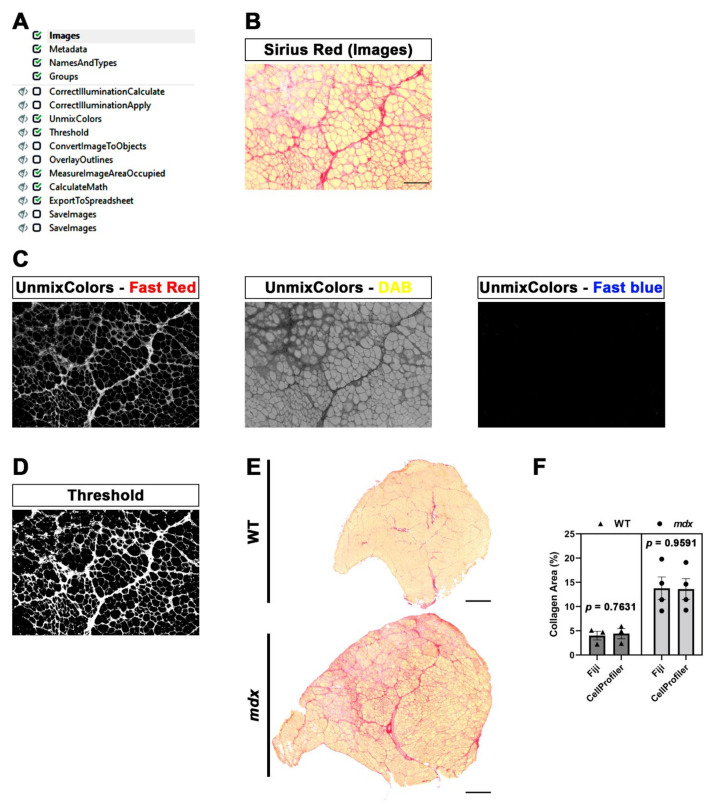


## 4. Discussion

Dystrophic muscles show high fiber heterogeneity, an elevated number of centrally nucleated fibers (CNFs) and heterogeneous cell populations and dynamics, especially in close proximity to injury sites. Moreover, due to massive extracellular matrix (ECM) deposition, significant interstitial spaces are detectable, especially at late stages of the disease [30]. For this reason, the full automation of fiber quantification can be tricky and inaccurate, and hence, a further step of manual adjustment and revision is required [15].

In the present study, by taking advantage of CellProfiler software, we developed two pipelines for the fully automated histological analysis of specific muscle hallmarks, starting from tibialis anterior (TA) sections of *mdx* mice, a widely used animal model for Duchenne muscular dystrophy (DMD), and their wild-type (WT) littermates. The quantifications performed with CellProfiler were then compared to those performed with Fiji in order to validate the robustness and efficiency of our method. It is therefore clear that this work did not aim to address differences between the two genotypes, which have been widely investigated in a multitude of previous works. We are interested in the validation of our CellProfiler-based pipelines. The first pipeline, named MyoProfiler, was designed for enhancing and redefining myofiber boundaries, starting from images of muscle cross-sections stained with laminin, α1 antibody; then, once the signal was satisfactorily identified, muscle sections were segmented, and their cross-sectional area (CSA) and minimum Feret diameter (MFD) were measured. The second part of MyoProfiler is aimed at identifying and segmenting DAPI-stained nuclei and at classifying segmented myofibers into CNFs and peripherally nucleated fibers (PNFs). The last part of the pipeline included the detection of macrophages, starting from the detection of F4/80, a pan-macrophage-specific surface marker. Finally, the SiriusProfiler pipeline was developed with the goal of identifying and quantifying ECM from Sirius red-stained muscle sections. To this end, we used the same samples used for the analysis by MyoProfiler. Concerning the time needed for analysis, Myoprofiler takes ~10 min to analyze 10 images, while an analysis performed with Fiji takes several hours for the user. Meanwhile, SiriusProfiler takes ~80 s to analyze 10 images, while manual quantification with Fiji takes ~13 min.

In the MyoProfiler pipeline, one of the most critical steps to deal with is the identification of muscle fibers and their appropriate segmentation. Muscle fiber integrity and laminin, α1 signals rely on the quality of muscle sections. Moreover, the laminin signal identifies not only myofiber boundaries but also nerve bundles, capillaries, veins, arteries and interstitial space (among fiber boundaries). It is therefore necessary to enhance the laminin signal, fix interrupted fibers where possible and reduce the background. All of the steps preceding the Threshold module resolve those problems. Moreover, we used the three-class Otsu method [27], in which we assigned the middle-class to the background in order to exclusively select the positive signal. Object segmentation (IdentifyPrimaryObjects module) was performed using the minimum-cross entropy threshold method [29]. Segmented fibers can be filtered out (FilterObjects module) if they do not achieve specified filter values (area, form factor and solidity). Finally, the automation of the procedure can be supported by a step of manual revision (EditObjects), necessary for an accurate evaluation of the quantification. Interestingly, the lack of this step was one of the limitations of the previously published MuscleAnalyzer pipeline [32].

Contrary to myofibers, the identification and segmentation of nuclei are easily affordable, as is nuclei segmentation from 2D cell culture images. This is possible since nuclei show a relatively uniform morphology, dimension and contrast due to the high contrast of the DAPI signal relative to the background. Once we reduced the salt-and-pepper background noise (MedianFilter) and increased the speckle features (EnhanceOrSuppressFeatures), we identified nuclei using an adaptive minimum-cross entropy threshold [29]. The quantification of nuclei worked appropriately, and the values obtained were comparable to those obtained with Fiji, although with Fiji, we detected slightly more nuclei. This is probably because in CellProfiler analysis, we excluded objects touching the borders. The identified nuclei were then shrunk to one point (ExpandOrShrinkObjects) and masked with 5-pixel-shrunken fibers (MaskObjects) in order to detect CNFs and PNFs with RelateObjects and ClassifyObjects modules. The strategy of identification and quantification of CNFs and PNFs, as with myofiber and nuclei, worked efficiently.

We also used the identified nuclei for the detection and quantification of resident (in WT mice) and infiltrating (in *mdx* mice) macrophages. Once the background noise had been reduced (MedianFilter) and the F4/80 signal had been enhanced (EnhanceOrSuppressFeatures), we once again used an adaptive minimum cross-entropy threshold for macrophage segmentation. Since F4/80 is a membrane surface marker, we chose to expand nuclei dimensions by 2 pixels and then expanded segmented nuclei with macrophages in order to better detect them. To finely select true macrophages, we decided to discard all objects that were partially masked (10% of masking region). Despite the challenge in detecting macrophages, especially in *mdx* cross-sections, we found overall comparable values in CellProfiler vs. Fiji analysis.

The SiriusProfiler pipeline, in contrast to MyoProfiler, works on brightfield images. This pipeline consists of a few crucial steps: the splitting of the input image into red, yellow and blue components and then the application of a Threshold module to the image corresponding to the red component. In this case, we used the three-class Otsu method [27], in which we assigned the middle-class intensity to the foreground, since Sirius red staining always shows a bright red signal and a less intense one. For Fiji analysis, we basically used the same approach, and we found no differences between the two methods, demonstrating the effectiveness of this method.

Occasionally, experimental errors can occur during the histological preparation of muscle samples, such as the wrong orientation of the sectioning angle (i.e., oblique sectioning), thus resulting in muscle fibers with a non-polygonal/non-circular aspect. This fact results in the incorrect measurement of muscle area by the CSA of myofibers, as has been previously demonstrated [33]. This inconvenient issue can be overcome using the minimal Feret diameter (MFD) as a parameter for the analysis of muscle fibers. Indeed, MFD is defined as the distance between the two parallel planes restricting the object perpendicular to that direction, so it is independent of the sectioning angle of the sample [33]. Using MyoProfiler, we quantified both CSA and MFD, but we present only the MFD quantification (Figure 3B). Moreover, isolated muscles can undergo poor inclusion or inappropriate storage before the sectioning. Finally, histological artifacts can occur during the sectioning of muscles, or they can have an uneven or irregular signal pattern due to errors occurring during the staining. Even one of these events can make the identification of objects and cellular components difficult, thus also affecting the quantification. This identification is allowed by CellProfiler thanks to the IdentifyPrimaryObjects module, which relies on a thresholding method that needs to be finely tuned in order to realize correct image segmentation. Fortunately, CellProfiler provides a test mode that makes it possible to test the pipeline on selected image sets and correct or change specific parameters once the output image has been generated. Of course, it is always better to test the pipeline, with a selected parameter, on a large set of images and on images of perfect or poorer quality, thus making the pipeline more robust. Finally, it is recommended to use appropriate conversion factors depending on camera properties and magnification, as well as parameters set in IdentifyPrimaryObjects, if necessary.

Undoubtedly, an automatic or semi-automatic approach should only be applied using good or average–good staining and images to avoid, for instance, the quantification of interstitial spaces or of other non-fiber structures. MyoProfiler proficiently quantifies muscle fibers with minimal error and ensures the possibility of an automated method to decrease the time required for quantification and, at the same time, offers a step of manual editing in order to maximize the efficiency and reliability of this approach, if needed.

In recent years, several other semi-automatic software packages or tools have been described [10,15,17,33]. Some of the issues with these approaches include the necessity of programming skills, the need for images of very high quality and the lack of implementation and of batch analysis. Moreover, Lau and colleagues proposed a method to detect and quantify muscle fibers and CNFs by using a previous version of CellProfiler. As also stated by the authors, their pipeline is not designed to identify specific cell populations and does not have a manual editing step [32]. Finally, Sanz and colleagues proposed a useful pipeline for the detection of fibers and the capillary-to-muscle fiber interface on muscle biopsies. Nevertheless, the pipeline has been designed using only muscles from healthy patients and images acquired with 20× magnification, thus reducing the reproducibility if input images are, for instance, 10× magnification-acquired images from dystrophic muscles [34]. Furthermore, the benefit of using CellProfiler for automated image analysis relies on its flexibility and the possibility of custom modifications that can also be applied by non-expert users. It can distinguish subtle changes and measure multiple properties at once. Moreover, it can perform batch analysis (thousands of images), and the latest version (CellProfiler 4) has been demonstrated to be faster and less tedious than previous ones [23]. Finally, the use of CellProfiler hints at the possibility to build user-friendly tools that are able to adapt and perform their tasks without needing to use long and more complex tools based on machine learning or deep learning.

The applicability of our pipelines relies on the possibility of also using them on images acquired from histological sections of human biopsies in order to obtain robust and valuable quantifications of histological parameters in both healthy and diseased patients (e.g., DMD patients). Indeed, as stated in Section A.4, our pipelines can be used also for quantifications on human muscle sections. Therefore, it would be possible, for instance, to histologically visualize the effect of corticosteroids, a widely used therapy in muscular dystrophies [35]. This should aid research and preclinical studies concerning muscle diseases.

To conclude, future directions starting from this work could include the development of novel CellProfiler-based pipelines aimed at quantifying other histological features of muscle histology as well as the detection and counting of other cell populations infiltrating or residing in skeletal muscle upon immunofluorescence for specific cell markers.

## 5. Conclusions

The CellProfiler-based pipelines designed in this study for the histopathological analysis of muscles allow the multi-parametric analysis of muscle sections in both physiological and pathological (DMD) conditions. These pipelines were designed in order to ensure automatic quantification of multiple images, starting from images acquired with non-automatic microscopes, and reduce the time usually spent on manual quantification. With this approach, it is possible to compare different experiments from different laboratories in a highly reproducible and easy-to-use interface. Finally, we developed a tool that should aid in the study and evaluation of pathologies affecting skeletal muscle by facilitating data generation and analysis, thus further improving the consistency of quantifications and the reliability of results.

## Data Availability

Not applicable.

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
