# Peer review of "A User-Friendly Approach for Routine Histopathological and Morphometric Analysis of Skeletal Muscle Using CellProfiler Software"

_diagnostics, 2022, doi:10.3390/diagnostics12030561_

Round 1

Reviewer 1 Report

“A user-friendly approach for routine histopathological and morphometric analysis of skeletal muscle using CellProfiler software”, submitted by Laghi et al., presents a method to automatically quantify multiple skeletal muscle histological features with the CellProfiler analysis software. The goal of the current manuscript is validation of the custom CellProfiler pipelines – MyoProfiler and SiriusProfiler. Strengths of the current manuscript is implementation of CellProfiler pipelines that are predominately automatic and based on free scientific analysis software. Weaknesses of the paper include the lack of detailed descriptions for optional manual steps and the lack a troubleshooting section. The target audience of the paper will likely enthusiastically welcome this new alternative for the unbiased analysis of pathophysiological studies of skeletal muscle disorders. Please see the comments below that aim to help improve the study and manuscript.

Major Concerns:

1. Although the pipelines are predominately automatic, semi-automatic options are mentioned in the manuscript. Detailing the optional manual editing steps would be very useful for readers that attempt to quantify muscle samples using MyoProfiler and/or SiriusProfiler.

2. The addition of a Trouble-Shooting section to the main text of the manuscript or as a supplemental section would be helpful.

Minor Concerns:

1. The anti-laminin antibody (L9393; Sigma-Aldrich) used in the current manuscript does not detect the alpha-2 subunit. The L9393 antibody detects the alpha-1 subunit. Please make the necessary corrections throughout the manuscript.

2. Challenges with macrophage detection were mentioned in the Results and Discussion. Do you envision that readers who plan to use the same pipeline to quantify satellite cells or other cell populations might run into similar challenges? Please address the possible issues and/or provide additional analysis and quantification, such as satellite cells, to confirm the versatility and possibly improve the confidence of those who plan to use MyoProfiler for spot detection. 

3. Fluorescently-conjugated wheat germ agglutinin (WGA), i.e., WGA-488, WGA-594, etc., has proven to be an effective alternative to picrosirius red in the assessment of extracellular matrix accumulation. Would it be possible to use MyoProfiler to analyze WGA fluorescent stains to quantify extracellular matrix?

4. Masson’s trichrome staining is often performed by researchers to evaluate the development of fibrosis. Is it possible to use SiriusProfiler to analyze Masson’s trichrome stained muscle samples? Or would the development of a new CellProfiler pipeline be necessary?

5. The directions for using the CellProfiler pipelines focus on the analysis of skeletal muscles from mice. However, many researchers who work with rats, pigs, dogs, or human subjects will attempt to use the approaches presented in the current manuscript. Consider which parameters will need to be altered. If possible, include a brief analysis of a muscle sample from one of the larger mammals (pig, dog, human) and emphasize the changes that need to be made when working through the steps during analysis.

6. Pre-clinical and clinical research assessing therapies aimed at restoration of dystrophin are well underway. A recent article by the Flanigan research group (PMID: 34847621) provides an automated analysis of dystrophin that has the potential to detect varying levels of dystrophin expression. Might a similar approach be possible with MyoProfiler? Limitations are understandable, but inclusion of a similar approach would improve the versatility of the MyoProfiler and excitement of the current manuscript.

Author Response

Response to Reviewer #1

Dear editor,

We wish to thank you for the editing of our manuscript entitled “A user-friendly approach for routine histopathological and morphometric analysis of skeletal muscle using CellProfiler software”.

The manuscript was revised in accordance with the reviewers’ requirements and comments. We also added the supplementary file named “Troubleshooting guide” outlining common issues that could arise during image analysis and how to fix them.

We hope that you and the reviewers will appreciate the work that we produced and we trust our work is now acceptable for publication in Diagnostics.

The changes introduced in the revised manuscript are described in the point-by-point letter below.

Reviewer #1

Major Concerns:

  1. Although the pipelines are predominately automatic, semi-automatic options are mentioned in the manuscript. Detailing the optional manual editing steps would be very useful for readers that attempt to quantify muscle samples using MyoProfiler and/or SiriusProfiler.

A:

We thank the reviewer for this suggestion and we agree about the importance of detailing the optional editing steps. Indeed, a detailed description of the optional manual editing step has been added in Paragraph 3.1 of Results section (Lines 270-282). Specifically, the SiriusProfiler pipeline does not include steps of manual editing because it detects positive areas and does not need to identify and segment objects. Conversely, regarding the MyoProfiler pipeline the additional manual editing step will allow for the manual screening and modification of the objects relative to the muscle fibers automatically detected. The EditObjectsManually module itself offers a “Help” function to explain the operation available to the user. Briefly, the manual editing allows to manually eliminate, create, split and re-shape the objects (muscle fibers).

  1. The addition of a Trouble-Shooting section to the main text of the manuscript or as a supplemental section would be helpful.

A:

We wish to thank the Reviewer for outlining the importance of adding a troubleshooting guide for users. As requested, we added as supplementary material, the troubleshooting guide (Troubleshooting_guide). Furthermore, we added the link to the Scientific Community Image Forum (https://forum.image.sc) in the troubleshooting guide. This forum is used by the community to help users with the realization, personalization and troubleshooting in performing image analysis, coherently with our aim to create an open cooperative toolbox.

Minor Concerns:

  1. The anti-laminin antibody (L9393; Sigma-Aldrich) used in the current manuscript does not detect the alpha-2 subunit. The L9393 antibody detects the alpha-1 subunit. Please make the necessary corrections throughout the manuscript.

A:

We would like to thank the Reviewer for pointing out this mistake. Opportune corrections have been made in the text, in figures 1 and 3 and relative figure captions.

  1. Challenges with macrophage detection were mentioned in the Results and Discussion. Do you envision that readers who plan to use the same pipeline to quantify satellite cells or other cell populations might run into similar challenges? Please address the possible issues and/or provide additional analysis and quantification, such as satellite cells, to confirm the versatility and possibly improve the confidence of those who plan to use MyoProfiler for spot detection.

A:

We would like to thank the Reviewer for addressing this crucial point. We realized that it is worth to integrate the manuscript with a paragraph focused on this point; to this end we included a dedicated paragraph in the Appendix section “Additional applications of MyoProfiler and SiriusProfiler pipelines” (Lines 701-727).

  1. Fluorescently-conjugated wheat germ agglutinin (WGA), i.e., WGA-488, WGA-594, etc., has proven to be an effective alternative to picrosirius red in the assessment of extracellular matrix accumulation. Would it be possible to use MyoProfiler to analyze WGA fluorescent stains to quantify extracellular matrix?

A:

As correctly stated by the Reviewer, fluorescently-conjugated WGA can be potentially used to detect extracellular matrix accumulation. We included in the Appendix section “Additional applications of MyoProfiler and SiriusProfiler pipelines” (Lines 729-741) detailed information about this potential application of the MyoProfiler Pipeline.

  1. Masson’s trichrome staining is often performed by researchers to evaluate the development of fibrosis. Is it possible to use SiriusProfiler to analyze Masson’s trichrome stained muscle samples? Or would the development of a new CellProfiler pipeline be necessary?

A:

We thank the Reviewer for rising this question. We have made several attempts with several images (used for another project carried out in our lab). We have found that Masson’s trichrome staining is successfully detected and quantified using the same threshold parameters used for SiriusProfiler. The details of this application are reported in the Appendix section “Additional applications of MyoProfiler and SiriusProfiler pipelines” (Lines 744-750).

  1. The directions for using the CellProfiler pipelines focus on the analysis of skeletal muscles from mice. However, many researchers who work with rats, pigs, dogs, or human subjects will attempt to use the approaches presented in the current manuscript. Consider which parameters will need to be altered. If possible, include a brief analysis of a muscle sample from one of the larger mammals (pig, dog, human) and emphasize the changes that need to be made when working through the steps during analysis.

A:

The Reviewer has pointed out an essential and impactful aspect of myobiology: the need to translate methods and research applied in mice to higher mammals and even human samples. Few considerations and suggestions about this point were included in the Appendix section “Additional applications of MyoProfiler and SiriusProfiler pipelines” (Lines 752-769).

  1. Pre-clinical and clinical research assessing therapies aimed at restoration of dystrophin are well underway. A recent article by the Flanigan research group (PMID: 34847621) provides an automated analysis of dystrophin that has the potential to detect varying levels of dystrophin expression. Might a similar approach be possible with MyoProfiler? Limitations are understandable, but inclusion of a similar approach would improve the versatility of the MyoProfiler and excitement of the current manuscript.

A:

We wish to thank the reviewer for pointing out this interesting biological question. In Flanigan et al., 2021 (PMID: 34847621), authors quantified, in muscle biopsies from DMD/BMD patients and healthy subjects, Dystrophin positivity and Dystrophin intensity. For Dystrophin positivity, they used Spectrin signal to locate muscle fiber perimeter. Then, they calculated the ratio between Dystrophin length (Fiber perimeter of Dystrophin+ signals) and Spectrin length (Fiber perimeter of Spectrin+ signals). For Dystrophin intensity, authors calculated the perimeter intensity for each object/myofiber. It is also possible to make those quantifications with CellProfiler. First of all, we would suggest to use Spectrin signal for the detection of myofibers, since it has a sarcolemmal localization as also Dystrophin has. Indeed, Laminin is located externally respect to sarcolemma. For this kind of analysis, the MyoProfiler pipeline is not suitable since all pre-processing steps for increasing Laminin signal and muscle boundary structures would alter the signal intensity of the protein. Indeed, we would suggest to use CellProfiler by performing a rough threshold (Threshold module) to Spectrin signal, in order to not artificially alter the original Spectrin signal, and then to invert the generated binary image (ImageMath module) for the following object segmentation (IdentifyPrimaryObjects) and filtering by size, form factor and circularity (FilterObjects) as we also did with the MyoProfiler pipeline. Then the per-object intensity can be calculated with the MeasureObjectIntensity module and then the Ratio between Dystrophin perimeter and Spectrin perimeter can be performed as authors did in Flanigan et al., 2021. Similarly, the perimeter intensity of Dystrophin intensity can be measured starting from object segmentation of Spectrin images. Unfortunately, we could not test this approach due to the lack of proper reagents available in the laboratory. For this reason, we will not mention this application in the manuscript.

ADDITIONAL CORRECTIONS

During revision of the paper, we have noticed and corrected an error in Figure 3B: in the X-axis of Fiber size distribution graph, we replaced “µm2” with “µm”.

Three references related to revised sections have been added.

Reviewer 2 Report

The authors have generated a novel platform for automated analysis of skeletal muscle sections using an adaptation of the open access CellProfiler software. The manuscript is well written, but requires some grammatical editing. The software is of significant interest to the skeletal muscle community and has been demonstrated by the authors to save significant time on histological analysis relative to manual analysis. It is useful that the authors have demonstrated the use of the software in both healthy and pathological (duchenne muscular dystrophy) muscle sections, showing that the software can be used accurately in both scenarios.

The major downfall of previously published automated analysis software is generally the ability to troubleshoot problems with the analysis when it arises. It would therefore be helpful if this manuscript included a supplemental troubleshooting file outlining common issues that could arise and how to fix them.

Author Response

Response to Reviewer #2

Dear editor,

We wish to thank you for the editing of our manuscript entitled “A user-friendly approach for routine histopathological and morphometric analysis of skeletal muscle using CellProfiler software”.

The manuscript was revised in accordance with the reviewers’ requirements and comments. We also added the supplementary file named “Troubleshooting guide” outlining common issues that could arise during image analysis and how to fix them.

We hope that you and the reviewers will appreciate the work that we produced and we trust our work is now acceptable for publication in Diagnostics.

The changes introduced in the revised manuscript are described in the point-by-point letter below.

Reviewer #2

The authors have generated a novel platform for automated analysis of skeletal muscle sections using an adaptation of the open access CellProfiler software. The manuscript is well written, but requires some grammatical editing. The software is of significant interest to the skeletal muscle community and has been demonstrated by the authors to save significant time on histological analysis relative to manual analysis. It is useful that the authors have demonstrated the use of the software in both healthy and pathological (duchenne muscular dystrophy) muscle sections, showing that the software can be used accurately in both scenarios.

The major downfall of previously published automated analysis software is generally the ability to troubleshoot problems with the analysis when it arises. It would therefore be helpful if this manuscript included a supplemental troubleshooting file outlining common issues that could arise and how to fix them.

A:

We wish to thank the Reviewer for outlining the importance of adding a troubleshooting guide for users. As requested, we added as supplementary material, the troubleshooting guide (Troubleshooting_guide). Furthermore, we added the link to the Scientific Community Image Forum (https://forum.image.sc) in the troubleshooting guide. This forum is used by the community to help users with the realization, personalization and troubleshooting in performing image analysis, coherently with our aim to create an open cooperative toolbox. Moreover, as requested by the Reviewer, we have performed grammatical editing.

ADDITIONAL CORRECTIONS

During revision of the paper, we have noticed and corrected an error in Figure 3B: in the X-axis of Fiber size distribution graph, we replaced “µm2” with “µm”.

Three references related to revised sections have been added.

Round 2

Reviewer 1 Report

All concerns have been adequately addressed. 

Only very minor editing needs to be done. For example, Line 760 in the Appendix, "must" should be replaced with "most".